# CRV-NET: Robust Intensity Recognition of Coronavirus in Lung Computerized Tomography Scan Images

**DOI:** 10.3390/diagnostics13101783

**Published:** 2023-05-18

**Authors:** Uzair Iqbal, Romil Imtiaz, Abdul Khader Jilani Saudagar, Khubaib Amjad Alam

**Affiliations:** 1Department of Artificial Intelligence and Data Science, National University of Computer and Emerging Sciences, Islamabad Campus, Islamabad 44000, Pakistan; 2Information and Communication Engineering, Northwestern Polytechnical University, Xi’an 710072, China; romilimtiaz@mail.nwpu.edu.cn; 3Information Systems Department, College of Computer and Information Sciences, Imam Mohammad Ibn Saud Islamic University (IMSIU), Riyadh 11432, Saudi Arabia; 4Department of Software Engineering, National University of Computer and Emerging Sciences, Islamabad Campus, Islamabad 44000, Pakistan; khubaib.amjad@nu.edu.pk

**Keywords:** deep learning, machine learning, computerized tomography, convolutional neural network, U-Net

## Abstract

The early diagnosis of infectious diseases is demanded by digital healthcare systems. Currently, the detection of the new coronavirus disease (COVID-19) is a major clinical requirement. For COVID-19 detection, deep learning models are used in various studies, but the robustness is still compromised. In recent years, deep learning models have increased in popularity in almost every area, particularly in medical image processing and analysis. The visualization of the human body’s internal structure is critical in medical analysis; many imaging techniques are in use to perform this job. A computerized tomography (CT) scan is one of them, and it has been generally used for the non-invasive observation of the human body. The development of an automatic segmentation method for lung CT scans showing COVID-19 can save experts time and can reduce human error. In this article, the CRV-NET is proposed for the robust detection of COVID-19 in lung CT scan images. A public dataset (SARS-CoV-2 CT Scan dataset), is used for the experimental work and customized according to the scenario of the proposed model. The proposed modified deep-learning-based U-Net model is trained on a custom dataset with 221 training images and their ground truth, which was labeled by an expert. The proposed model is tested on 100 test images, and the results show that the model segments COVID-19 with a satisfactory level of accuracy. Moreover, the comparison of the proposed CRV-NET with different state-of-the-art convolutional neural network models (CNNs), including the U-Net Model, shows better results in terms of accuracy (96.67%) and robustness (low epoch value in detection and the smallest training data size).

## 1. Introduction

The detection of infectious diseases is a continuous job for modern digital healthcare systems. The report of coronavirus (COVID-19) was in the city of Wuhan in China, and it was initially discovered in the transmission from animals to humans [1]. COVID-19 is a deadly virus that has killed hundreds of people every day across the globe since December 2019 [2]. It is a communicable disease (transferable from one human body to another human body), and the discovery of robust methods to diagnosis its intensity is still open to different scientist’s communities. At the first stage, nucleic acid amplification tests (NAATs), antigen tests and self-tests are normally in practice for diagnostic purposes, but intelligent diagnostic tools are still on the way for robust COVID-19 intensity measurement [3]. As a way to ensure human survival, one of the most successful strategies for infection tree trimming is the early detection of this illness [2,3]. The increasing number of COVID-19 patients is overwhelming in many countries’ healthcare systems. As a result, a reliable automated methodology for recognizing and measuring infected lung areas would be extremely beneficial. From a clinical perspective, radiologists have detected three forms of COVID-19-related abnormalities in CT images of lungs that used ground glass opacification (GGO), consolidation and pleural effusion [2,4].

A number of computational techniques are used to recognize different human diseases by using medical images of the lungs and chest [2,3]. For the accurate recognition of anomalies in medical images, deep learning models have highlighted a standard strategy to build the network’s capacity for the simulation of higher-order records. Similarly, tumors (in the infected area) are the clear target of deep-learning-based medical segmentation systems. From one instance in [5], the authors attempted to assess a tumor in the brain with MRI images by using the combination of U-Net along with SegNet, achieving a 0.99 accuracy. Similarly, [6] proposed DL structures that were used to develop a lung cancer screening tool to reduce the false positive rate in CT scan images. CT scan images are also used for breast tumors detection with image segmentation techniques [7], with a mean accuracy of 0.90 with generative adversarial networks (GANs) and convolutional neural networks (CNNs). For the robust diagnosis of human diseases, different researchers attempted to segment the medical image by highlighting human body parts, such as kidneys, with images segmented in [8]; the lungs, with images segmented in [9,10]; the liver, with images segmented in [11]; the brain, with images segmented in [12,13]; and temporal bones, with images segmented in [14].

The literature further delivers the various clinical decision support platforms, which used the DL-based medical image segmentation technique very efficiently in, for example, breast cancer classification [15,16]. Similarly, few studies provide a better solution for patch classification in CT scan images [17,18]. Along with the diagnosis of different diseases via machine learning techniques, many research initiatives have also been carried out for the identification of COVID-19 by employing deep learning models in X-Ray and CT scan images and obtaining substantial findings [19,20]. For instance, in [18], a stacked sparse autoencoder (SSAE) model for the efficient segmentation of CT scans with the identification of foreground and background patches was conducted. Furthermore, in [20], extensive experiments were performed on CT scan images for the accurate classification of COVID-19 through the encoder–decoder convolution neural networks (ED-CNNs), U-Net, and the Feature Pyramid Network (FPN). Conversely, for the early diagnosis of COVID-19, semantically segmenting X-Ray and CT scans is still an open window for medical computational research communities [21,22].

Different computational researchers have explored a variety of DL structures to detect COVID-19-infected individuals using medical images of X-Ray and CT scans. For the accurate detection of COVID-19, one study used a CNN by using X-Ray images as an input to create a binary classifier [23] and a multi-class classifier, achieving a 0.98 accuracy in binary and a 0.87 accuracy in multi-class sections [24,25]. Similarly, another research work used the Xception and ResNet50V2 models to recognize the COVID-19 cases in CT images, with 0.99 accuracy [26]. Furthermore, recent studies [27,28,29] employed different DL structures, especially CNN, on medical images (CT Scan and X-Rays) and found accuracy scores ranging from 83% to 98%.

Recently, a few other attempts to identify COVID-19 cases in the segmenting of medical images of COVID-19 patients have been reported by using X-Rays and CT scans [30,31]. Implementing semantically segmentations would be a critical component of any healthcare system that prioritizes the patients based on severity [32]. Along with the patient’s condition, it would diagnose the infection and its intensity early. In one instance [32], a deep CNN was used as a binary segment, and it was compared to different U-Net structures. In another instance [32], the authors obtained a 0.73 dice value, 0.75 sensitivity, and 0.73 precision value for the detection of COVID-19. Another study [24] highlights different applications for a binary segmentation tool for the detection of COVID-19 and obtained fair efficiency parameters including a Dice value of 78%, a precision of 86%, and a sensitivity of 94%. Using binary segmentation tools, the authors in [33] used a U-Net along with a fully convolutional network model (FCN); their attempt fared well in terms of efficiency parameters (precision and accuracy) but not so well in Dice parameters. Moreover, Ref. [34] describes the creation of two unique deep neural network structures (Inf-Net and Semi-Inf-Net) for segmenting infected areas, as well as segmenting GGO and consolidation.

One of the most serious difficulties in COVID-19 diagnosing difficulties is the lack of synchronized patterns in data [35,36]. Another constraint of the pattern recognition of COVID-19 is the restricted availability of specialists to segment CT images [37]. An increasing number of experts can validate the accuracy of the retrieved lesions’ boundaries, which might eventually lead to an improvement in model accuracy. Therefore, in the presence of these hardware constraints, it is also difficult to test and analyze the effect of altering various parameters and hyperparameters on a neural model’s performance [38].

In this article, we proposed a new and robust method for the detection of COVID-19 using limited data from lung CT scan images. The customized limited dataset of 321 CT scan images is prepared using the public SARS-CoV-2 CT Scan dataset [39] after the pre-processing stage. The limited customized dataset includes 221 CT scan ground truth images and 100 CT images. The primary objective of this study is to build a model that is well trained for the robust accurate detection of COVID-19 images by using limited data sources. The major contributions of this research work are summarized as follows:This research work delivered the accurate detection and automated segmentation of COVID-19 infected areas in lung CT scan images.The proposed CRV-NET architecture is the modified version of generic U-Net architecture that has included the weight pruning approach on both encoder and decoder sides for robust COVID-19 detection using limited data sources of lung CT scan images.The proposed CRV-NET architecture is compared to different state-of-the-art variants of U-Net architectures in terms of accuracy and robustness (least time complexity) for COVID-19 detection.This study delivers the future concept of federated learning-based CRV-NET for robust COVID-19 detection in intra-patient hospital cases.

The rest of the article is organized into a few sections. Section 2 highlights the methodology, which is further classified into four subsections: Section 2.1 belongs to deep CNN based classification, Section 2.2 highlights the general flow of the pre-processing stage, and Section 2.3 covers the operation of generic U-Net architecture in terms of the detection of COVID-19, and Section 2.4 discusses the COVID-19 detection workflow using the proposed architecture, which is a modified version of generic U-architecture. The Results are thoroughly discussed in Section 3 and highlight the significance of the proposed architecture in the context of effectiveness and robustness. Furthermore, state-of-the-art comparisons are also performed in Section 3. The robustness of the proposed architecture is highlighted in Section 4, Discussion. Finally, Section 5 highlighted the conclusion of this study.

## 2. Materials and Methods

This study proposed a unique method for the accurate and robust detection of COVID-19. A proposed CRV-NET, generic U-Net and CNN classifier are used to detect and segment images of COVID-19. According to Figure 1, a public dataset, SARS-CoV-2 CT Scan dataset is used to formulate the customized dataset. The customized dataset is a combination of ground truth images after pre-processing phase (labeled by radiologists) and general lung CT scan images. In the pre-processed stage, binary operations are performed for the removal of unnecessary minor details in ground truth CT scan images. According to Figure 1, the unlabeled CT scan images and ground truth CT scan images are processed in two different paths. At first, the CT scan images of the public dataset are processed by a well-tuned CNN model for classifying COVID-19. Concurrently, after pre-processing stage, the 221 CT scan ground truth images and 100 raw CT scan images of customized dataset are employed for training and testing with different architectures of U-Net (U-Net, U-Net++ and proposed CRV-NET). Thus, after the classification of COVID-19 by the well-tuned CNN model, these classified images are processed using U-Net, U-Net++ and proposed CRV-NET for accurate and robust segmentation of COVID-19 to measure the intensity of the virus.

Figure 1 highlights the work scheme for robust and accurate detection of COVID-19. It is a composition of two main parts, namely, the CNN-based classification and pre-processing stage along with image segmentation using different architectures of U-Net.

### 2.1. Deep-CNN-Based Classification

In the classification phase of the proposed method, a CNN-based classification delivers the efficient and robust detection of COVID-19. In the proposed method, a deep convolutional neural network model is employed that contains 16 layers. Furthermore, the structure of CNN with kernel size 3 × 3 and filters 16, 32, 64, 128 and 256 are employed for classification. Figure 2 presents the layer structure of the CNN model that is used in the classification of COVID-19.

According to Figure 2 above, the structure of deep CNN is highlighted as first using the 3 convolution layers, a 3 × 3 kernel and 16 filters. After that, a max pool layer with 2 × 2 strides, 3 more convolution layers, 3 × 3 kernel and 32 filters are employed. Further in the depth of the structure, add another max pool layer with strides 2 × 2 and 3 more convolution layers along with 128 filters. Similarly, the final max pool layer and the final convolution layer with 256 filters are further connected with 2 hidden layers with 230,400 and 460,800 perceptrons. Lastly, the two output layers are classified by the existence or absence of COVID-19 in CT scan images.

### 2.2. Pre-Processing

In pre-processing stage, according to Figure 1, the ground truth images are labeled by radiologists. At the start of this stage, set the threshold point with the help of rescaling intensity, which ranges from 92 to 98. Next, assign the binary operations, including the binary opening (0.5 > set 1 value) and the binary closing (0.5 < set 0 value), and remove small objects with dice range of 2 to 4 from the defined image. The defined setting of the pre-processing stage plays a vital role in the next two-way classification for the identification of the infected area (the detection of COVID-19) in a CT scan of the lungs.

After the pre-processing phase, the customized small-scale dataset of 331 images is employed for image segmentation to highlight the infected areas in CT scan images. In the segmentation phase, the COVID-19 classified images are processed with the proposed CRV-NET architecture and generic U-Net for segmentation of the COVID-19 infected areas to highlight the intensity of the virus. The combination of proposed CRV-NET, generic U-Net, and U-Net++ are used to validate the segmentation results of COVID-19 in CT scan images.

### 2.3. U-Net Architecture

U-Net is a CNN-based architecture that plays a vital role in tumor detection in MRI and CT scan images [40,41]. In U-Net, performing the image segmentation for detection of the infected area without losing the feature mapping is the most significant part. For semantic segmentation, especially in the CT scan and MRI images, different U-Net structures are used [5,40]. CNN in U-Net is focused on learning representations of data and hierarchical feature learning. For feature extraction, CNN employs an arrangement of several layers of nonlinear processing identities, and the output of each sequential layer becomes the input of the next one, which aids in data abstraction. The standard U-Net design is shown in Figure 3 [41].

The standard U-Net design in Figure 3 highlights the dense structure. The denser part of U-Net is the 32 × 32 pixels (lowest resolution) along with a multi-channel function map that is represented in the blue box. Furthermore, the number of channels is shown on the box’s end, white boxes show the feature maps that have been copied and arrows represent the upsampling and downsampling operations.

### 2.4. Proposed CRV-NET Architecture

The proposed CRV-Net is the modified version of generic U-Net, which works on limited data with a fair accuracy level. In CRV-NET, the CNN model is operated in upsampling and downsampling operations in a similar way to U-Net. The reduction in computational cost in terms of minimum time consumption and accurate detection of COVID-19 is the primary objective of the proposed CRV-NET. Therefore, the CRV-NET increases the level of COV and pooling layers by using up to 2048 filters. Additionally, the weight dropout scheme follows in each operational phase of CRV-NET. Figure 3 shows the structural view of the proposed CRV-NET.

CRV-NET architecture is divided into two state-of-the-art parts, namely, downsampling and upsampling. In Figure 4, the left side highlights the downsampling and is known as a contracting path. The image size continues to reduce in downsampling with the filter’s increment. The number of features would be doubled every time in the contracting direction. The process of downsampling is accomplished by repeating two 3 × 3 convolutions with the usage of rectified linear unit (ReLU) activation function and max-pooling by setting the kernel size of 2 × 2 and a stretch of 2. Conversely, the upsampling represented on the right side of Figure 3 is also known as an expanding path. In upsampling, the size of the image is gradually enhanced by decreasing the size of the filter. The same configuration is used in the upsampling process as was used in the downsampling, namely, two 3 × 3 convolutions, followed by a ReLU activation function, except that the layers in the expansive path are concatenated with the corresponding layers in the contracting path. At the final stage, a 1 × 1 convolution is used in the last layer to map with a desired class.

The CRV-NET architecture was trained on 321 images with the ground truth images and minimum epoch values that reflected the least execution time (computation cost). This optimal training process of CRV-NET plays a vital role in the robust detection of COVID-19. In addition, the padding is equal to that of the input image in order to resize the image after upsampling.

Moreover, Table 1 highlighted the hyperparameters list used in training the proposed CRV-NET. In CRV-NET, Adam is used, which is the replacement optimization algorithm for stochastic gradient descent for training CNN models. Therefore, CRV-NET is the preferable architecture for the robust detection of COVID-19 in limited data sources.

Along with accuracy, the short computational time of CRV-NET is the key parameter in the diagnosis of COVID-19. In CRV-NET, the weight pruning policy in each stage of upsampling (encoder), downsampling (decoder) and dense convolution layers plays a key role in the optimization of accurate results. Figure 5 presents the component’s view of CRV-NET.

For performance improvement of convolution neural structures, in terms of the combination of activation functions, the input layer uses the ReLU activation function, and the output layer uses the sigmoid function. The reason for using the ReLU function over other activation functions is that it does not activate all the neurons at the same time. Equation (1) shows the standard discrete nature of the ReLU activation function:(1)$$f\left(x\right)=max(0,x)$$

Similarly, at the output layer, the sigmoid function is used to perform the binary classification. The virus-infected area is quickly identified via binary classification. The range of the sigmoid function is 0.0 to 1.0, and Equation (2) shows the non-linear nature of the sigmoid function:(2)$$f\left(x\right)=\frac{1}{1+e^{-\left(x\right)}}$$

In the CRV-NET, class 0 belongs to the downsample because the majority of the pixels belong to it, and it does little to boost distinguishing ability. In the CRV-NET, only select the slices in the image that contain infected-area pixels in the downsampling process. As discussed above, the downsampling process continuously shrinks the image size and increases the size of the filter to 2048. Furthermore, in each step of downsampling, the image size reduces by up to half when we are applying filters.

Similarly, the upsampling process is executed in a reverse way, such as reducing the number of filters by enlarging the image size and maintaining a connection with downsampling values to obtain features. At the last layer, the final output image was converted into a binary image, and the sigmoid function was used to obtain a more accurate value of COVID-19 detection.

## 3. Results

To evaluate the performance of the proposed CRV-NET, efficiency parameters, including mean accuracy, sensitivity and the epoch ratio, are calculated along with the dice score. A dice score is computed for the identification of the feature recognition of the infected area. In feature recognition, the dice score coefficient is used to measure the accuracy between the ground truth images and segmented images. The dice coefficient is a fixed similarity measure function that is typically used to determine the similarity of two samples (ground truth images and segmented images), with a value range of [0, 1]. Furthermore, dice loss will be smaller in the dice coefficient in the case of two samples that highlight a high similarity index.
(3)$$Dice\;Loss\left(A,B\right)=1-\frac{2\left|A\cap B\right|}{\left|A\right|+\left|B\right|}$$

Equation (3) highlights the standard form of dice loss in which set *A* belongs to the ground truth images and set *B* represents the segmented images. The dice loss ratio in Equation (3) highlights the ratio of feature loss between the ground truth images of CT scans and the segmented images of CT scans. *Dice loss* is the efficiency measurement gauge of CRV-NET. The minimum ratio of dice loss highlights the least feature mapping loss between the ground truth images and the segmented images. Conversely, if the dice loss ratio is large, it reflects a huge feature mapping loss.

Additionally, to cross-validate the performance of the proposed CRV-NET, it is recommended to measure the statistical parameters, including *TP* (true positive), *TN* (true negative), *FP* (false positive) and *FN* (false negative), and then, with the help of these parameters, compute the efficiency gages in terms of specificity (*Sp*), sensitivity (*Se*) and accuracy (*Acc*). Equation (4) represents the specificity that delivers the correct analysis of a person not having a tumor:(4)$$Sp\left(\%\right)=\frac{TN}{TN+FP}\times 100$$

Similarly, Equation (5) highlights the sensitivity (*Se*) that highlights the correct analysis of a person having a disease:(5)$$Se\left(\%\right)=\frac{TP}{TP+FN}\times 100$$

Equation (6) represents the accuracy factor (*Acc*) of the proposed method in terms of accurate classification:(6)$$Acc\left(\%\right)=\frac{TP+TN}{TP+TN+FP+FN}\times 100$$

The proposed CRV-NET architecture was validated on the limited dataset (321 images) and the minimum epoch ratio. The proposed architecture obtained an accuracy of 96.22%, which is better than the state-of-the-art generic U-net, and the least computation complexity in comparison (less computation cost in terms of execution time). Therefore, according to Figure 1, the second track of the work scheme is executed that contains the generic U-Net architecture and proposed CRV-NET architecture.

According to Figure 1, after the pre-processing stage on the dataset, the execution of the generic U-Net and proposed CRV-NET architectures are operated on lung CT scan images and ground truth images. The effectiveness of generic U-NET architecture and the proposed CRV-NET architecture is validated by efficiency gages, including the epoch ratio, the accuracy of COVID-19 detection, specificity and sensitivity. The robustness factor in the context of computational cost is also a core parameter in the accurate detection of COVID-19. Hence, the epoch ratio is measured as a computational cost parameter for COVID-19 detection. Figure 6 presents the smallest epoch ratio of the proposed CRV-NET in terms of accuracy and loss for the detection of COVID-19.

Figure 6 shows the incremental variation of the CRV-NET epoch ratio. Figure 6a,b highlight the accuracy and loss parameters of CRV-NET in the COVID-19 detection process. Figure 6a presents the least increment variation in epoch values, enhancing the accuracy ratio up to 96% with limited input data, which means the model is well trained in CRV-NET architecture. Similarly, Figure 6b highlights the gradual decline of the loss parameter with a similar pattern of the epoch incremental ratio. Figure 6b also represents the model that is fine tuned in CRV-NET. Figure 7 represents the findings for the robust detection of COVID-19 via segmentation and detection.

In a sample of four images, Figure 7a presents the CT scan images, which are segmented and masked in Figure 7b after the implementation of image processing techniques, including the removal of small objects, the binary opening and the binary closing. Finally, Figure 7c shows the detection of COVID-19 in the yellow part of the CT scan images. According to Figure 7, image 3 shows the worst dice score after the execution of CRV-NET because the ground truth does not match accurately with the detected result. However, the proposed CRV-NET images 3 and image 4 also highlight the segments of infected areas that are not present in the ground truth image. Similarly, Figure 8 presents the dice segmentation result with the bar chart.

Figure 8 shows the dice similarity index on four COVID-19 detected images. The dice similarity index covers the exact ratio of COVID-19 infected areas in images. Figure 8 shows the 64.45% dice similarity index of COVID-19 detection in image 3, which is easily observed in the third section of Figure 8. Similarly, images 2 and image 4 represent the 63.38% dice score, and image 1 highlights the 61.21% dice score.

## 4. Discussion

In operations of the proposed work scheme, the trained model accuracy of the generic U-Net is 95.81% using the ground truth CT scan image dataset, and the validation accuracy using the CT scan image dataset is 94.50%. In the proposed CRV-NET, the accuracy has been increased with the trained model accuracy reaching up to 96.67% by using the ground truth CT scan image dataset, and the validation accuracy is 96% when using CT scan images dataset. The first level of comparison highlights that a CRV-NET is a step higher than the generic U-Net in terms of accuracy. For the cross-validation of the proposed CRV-NET architecture, the state-of-the-art comparison is performed in Table 2. For the state-of-the-art comparison, different efficiency parameters, including sensitivity, accuracy, epoch value and the dice coefficient, are considered.

Table 2 highlights the larger dataset of lung CT scan images in previous methods, showing that the current dice score and specificity parameters are high for measuring the intensity of COVID-19 detection compared to the accuracy and sensitivity parameters of previous medical imaging diagnostic studies [47,48]. The proposed CRV-NET dominates in terms of efficiency parameters such as the accuracy of COVID-19 detection, sensitivity and robustness (low epoch value). According to the target of COVID-19 detection in the limited data source, the high accuracy and sensitivity percentage in the limited dataset size highlight that the proposed CRV-NET is better than previous U-Net architecture. Additionally, the small epoch value of the proposed CRV-NET highlights the significance of the low computation cost compared to previous studies that used different U-Net architecture.

Furthermore, U-Net++ is another refactored version of U-Net architecture that delivers optimized accurate results in CT scan images for the diagnosis of different diseases [41,49] U-Net++ is a composition of nest U-Nets and operates with node pruning policy via nested and dense skip connections between upsampling and downsampling. The effectiveness of U-Net++ is validated in the test environment of CRV-NET. Table 3 shows the performance comparison of CRV-NET and standard U-Net++ using the customized dataset of 331 CT scan images.

According to Table 3, the customized dataset of 331 CT scan images is operated on both U-Net++ and CRV-NET architectures. In CRV-NET, the minimum epoch’s increment generates better training data accuracy and testing data accuracy along with minimum loss values. Conversely, in U-Net++, large incremental variation in epochs delivers the low training and testing accuracies compared to CRV-NET.

Nonetheless, the accurate early diagnosis of COVID-19 is the primary objective of modern healthcare systems, but the transformation of such early diagnoses using wearable gages is the future vision. To concatenate the future vision, the integration of federated learning and CRV-NET is needed for the robust recognition of COVID-19 via wearable gages. Figure 9 presents the unique work scheme of federated learning for the robust detection of COVID-19 that will be implemented in the future.

A trustworthy federated learning structure will be used in the future for the detection of COVID-19 with the integration of CRV-NET [50,51]. According to Figure 9, real-time data streams that are fetched from standard wearable gages [52,53] are trained on local CRV-NET, which is further tested and validated by a cloud-based Master CRV-NET and an active repository data center [54,55]. The robustness and accurate factors of CRV-NET will help the early diagnosis [56,57] of COVID-19 via federated learning, which is highlighted in Figure 9.

## 5. Conclusions

The improvement of accuracy and robustness in detecting COVID-19 is the top clinical requirement following the emergence of this new global virus. Previous studies have delivered a fair contribution to the diagnosis of COVID-19, but the robustness factor is compromised. This research work proposed a unique structure of U-Net in terms of CRV-Net for the accurate and early diagnosis of COVID-19 with the least computational cost (the smallest epoch ratio). The proposed CRV-NET architecture worked for the robust detection of COVID-19, which is a refactored version of generic U-Net architecture. A public SARS-CoV-2 CT scan dataset is used for experimental purposes and the customized dataset. The customized dataset is used for training the model with 331 ground truth images and 100 images. The proposed CRV-NET’s accuracy, 96.67%, is better than that of the generic U-Net architecture. Similarly, the epoch value of CRV-NET is seven, which is quite impressive and supportable for robustness (least computational cost). Moreover, wearable gages will be used in the future for the early diagnosis of COVID-19 by using the trustworthy federated learning work scheme with an embedded part for CRV-NET.

## Figures and Tables

**Figure 1 diagnostics-13-01783-f001:**
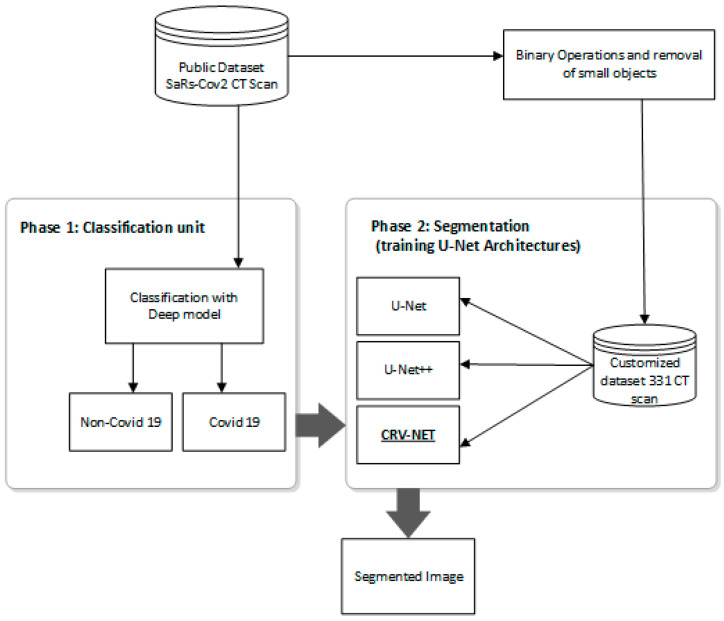
Proposed workflow for COVID-19 detection and intensity measurement.

**Figure 2 diagnostics-13-01783-f002:**
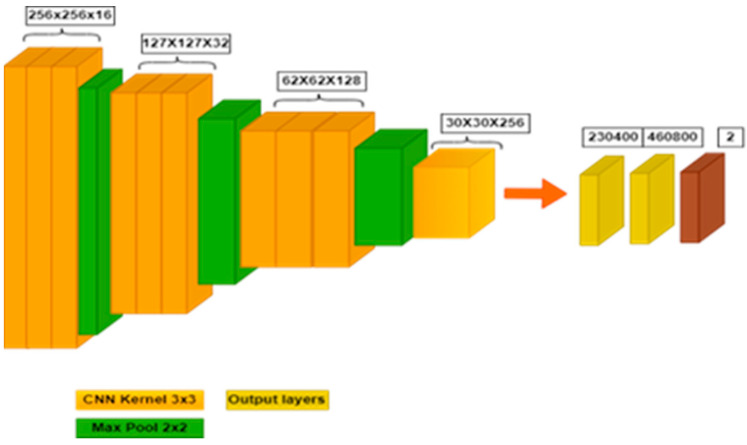
Layered view of deep convolution neural network.

**Figure 3 diagnostics-13-01783-f003:**
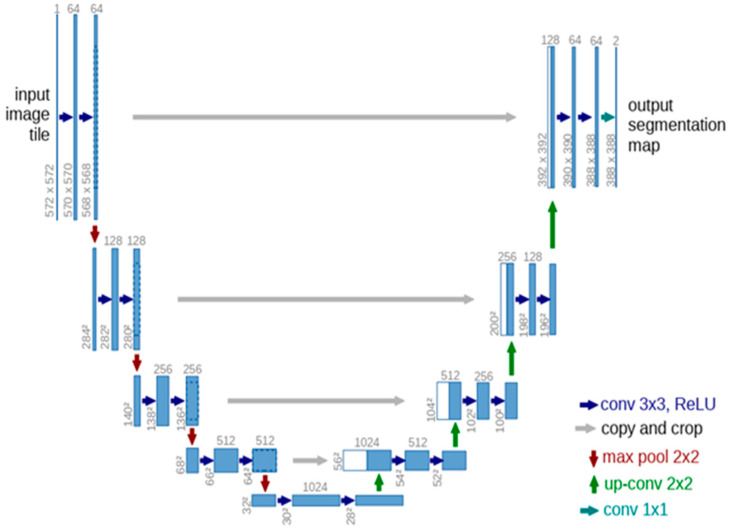
Generic U-Net Architecture [41].

**Figure 4 diagnostics-13-01783-f004:**
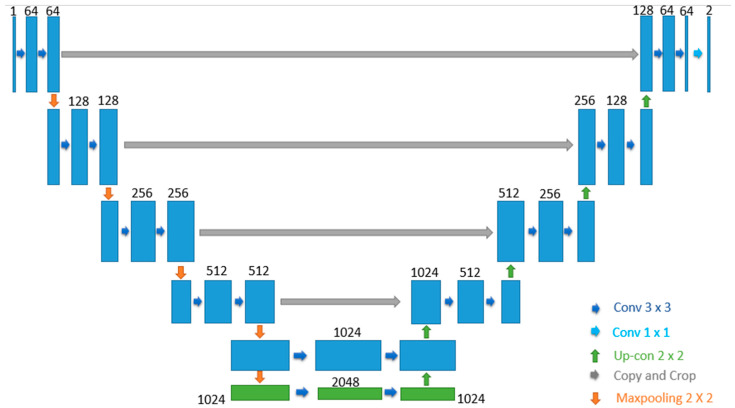
Proposed CRV-NET architecture.

**Figure 5 diagnostics-13-01783-f005:**
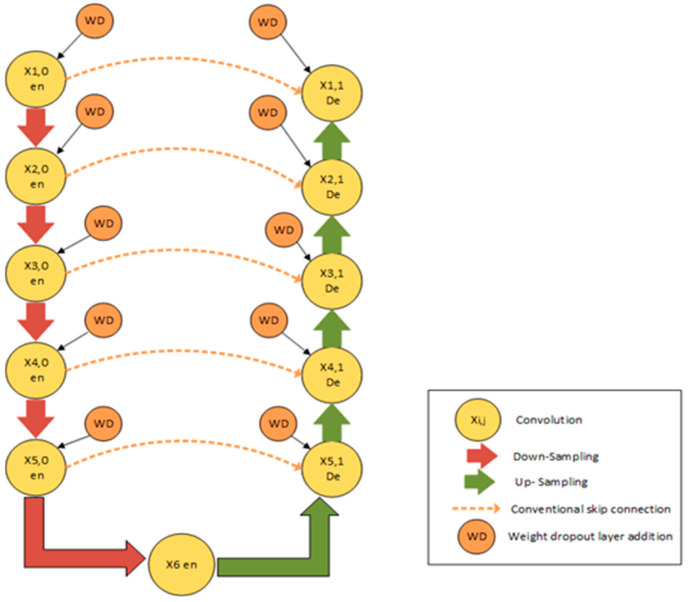
Basic components of CRV-NET architecture.

**Figure 6 diagnostics-13-01783-f006:**
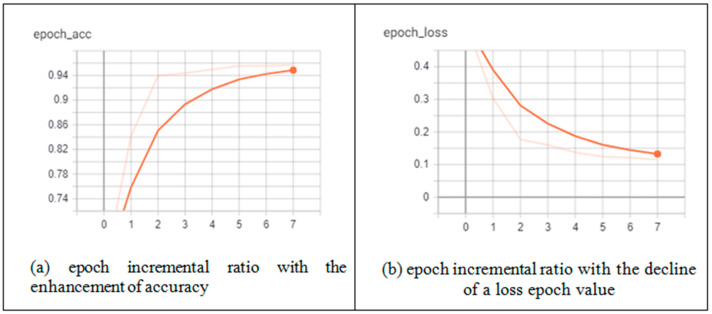
Epoch incremental variation in accuracy and loss parameters of CRV-NET.

**Figure 7 diagnostics-13-01783-f007:**
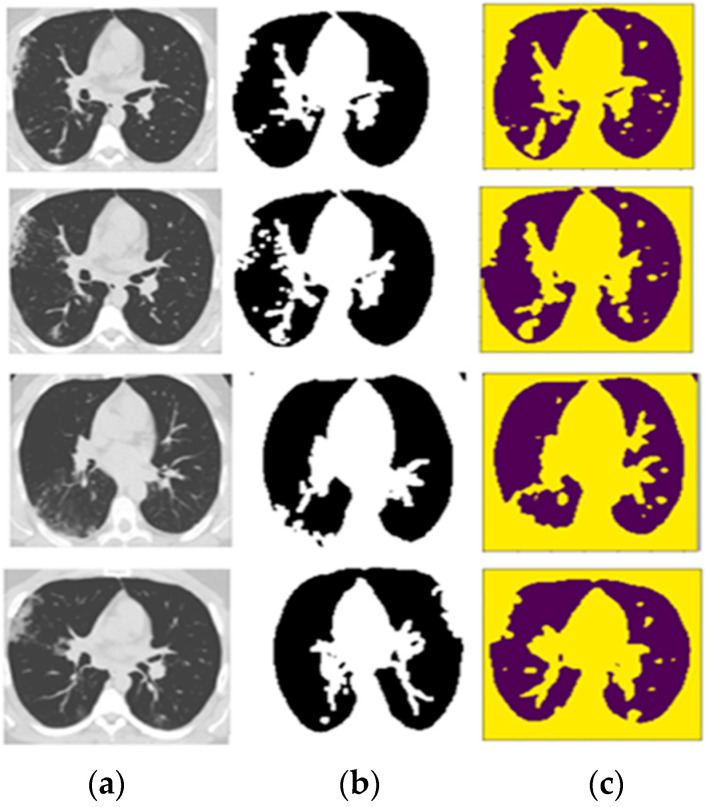
Detection of COVID-19 using CRV-NET. (**a**) Raw CT scan images of the public dataset; (**b**) raw CT scan images after image processing is implemented in the context of the removal of small objects, the binary opening, and the binary closing; (**c**) raw CT scan images with the yellow region highlighting the detected area of COVID-19.

**Figure 8 diagnostics-13-01783-f008:**
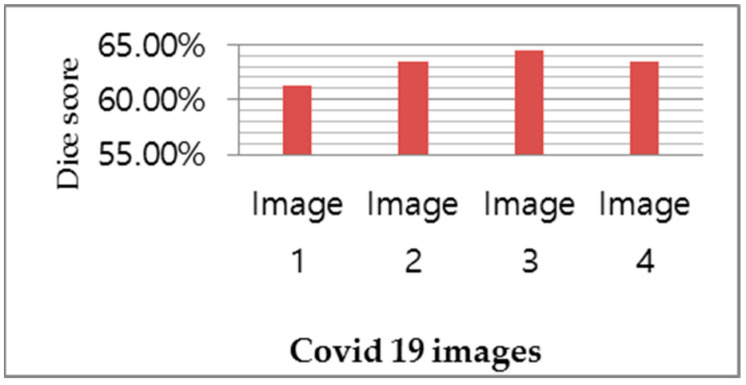
Dice loss ratio of COVID-19 detected images.

**Figure 9 diagnostics-13-01783-f009:**
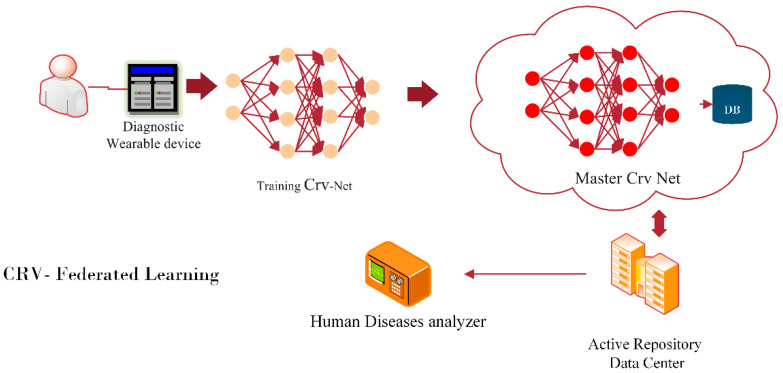
Future robust COVID-19 prediction wearable gages with embedded CRV-NET federated learning.

**Table 1 diagnostics-13-01783-t001:** Hyper-parameter values used in model training (CRV-NET and U-Net).

CRV-NET Hyper-Parameters	
Batch size	16
Train Validation Test split	0.7, 0.1, 0.2
Optimizer	Adam
Input shape	128 × 128 × 3
Dropout	0.2
Zoom range	0.2
U-Net Hyper-parameters	
Batch size	64
Train Validation Test split	0.8, 0.1, 0.1
Optimizer	Adam
Input shape	512 × 512 × 1

**Table 2 diagnostics-13-01783-t002:** State-of-the-art comparison of proposed CRV-NET.

Ref	Method	Dice	Dataset Size	Sensitivity	Specificity	Accuracy	Epoch
[42]	U-NET	83.10%	473 CT scan images	86.70%	99.00%		50
[40]	SegNet	74.90%	21,658 CT scan images	94.50%	95.40%	95%	160
[43]	COVID-NET		13,975 CT scan images	80%		93.3%	22
[44]	U-Net	92%	639 CT scan images	81.8%	95.2%	94.0%	100
[45]	3D U-Net	76.1%	20 CT scan images	95.56%	99.8%	95.56%	312
[46]	CovidDenseNet		4173 CT scan images	86.14%	95.46%	95.76%	150
**Purposed Method**	**CRV-NET**	**64.45%**	**331 CT scan images**	**96.67%**	**90%**	**96.67%**	**7**

**Table 3 diagnostics-13-01783-t003:** CRV-NET compared to U-Net++ using the customized dataset.

Epoch	Training Loss		Training Accuracy		Test Loss		Test Accuracy	
	U-Net++	CRV-NET	U-Net++	CRV-NET	U-Net++	CRV-NET	U-Net++	CRV-NET
1	0.6006	0.5569	0.3994	0.4431	0.8247	0.5017	0.1753	0.4983
2	0.4745	0.3036	0.5255	0.6964	0.784	0.4242	0.216	0.5758
3	0.3936	0.2046	0.6064	0.7954	0.7414	0.2599	0.2586	0.7401
4	0.3452	0.1544	0.6548	0.8456	0.7159	0.2522	0.2841	0.7478
5	0.3101	0.1427	0.6899	0.8573	0.6527	0.1751	0.3473	0.8249
6	0.2841	0.1338	0.7159	0.8662	0.5596	0.1705	0.4404	0.8295
7	0.2642	0.1175	0.7358	0.966	0.4401	0.236	0.5599	0.8974

## Data Availability

All the datasets used in this study are publicly available; no exclusive dataset is used in this study. Readers can extend this research for the diagnosis of different communicable diseases using the structure of CRV-NET and reproduce the experimental setup of CRV-NET to measure the intensity of COVID-19 by using the following link: https://github.com/UZAIRIQBAL-dev/CRV-NET-.git (accessed on 4 June 2022).

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
