# Peer review of "CRV-NET: Robust Intensity Recognition of Coronavirus in Lung Computerized Tomography Scan Images"

_diagnostics, 2023, doi:10.3390/diagnostics13101783_

Round 1

Reviewer 1 Report

The research is good and needs some processing:

- The introduction should be more clear, as it contains an introduction to Covid-19, its causes, spread, danger, and treatment. Then he discusses the techniques used to discover it. After that, the shortcomings of manual diagnosis and the importance of artificial intelligence to address this shortcoming.

- Figure 1 is not clear. It should be more clear for easy understanding of the working of the manuscript.

- The title of the research is not identical to Figure 1. Where Figure 1 is the final step is segmentation, not Recognition or classification. How do you explain this?

Schedule 3 must come before Schedule 2.

- The authors mentioned in Table 3 the accuracy of their system is 96.67%, while in Table 2, which displays the results of the CRV-NET system, I did not find the same accuracy that they mentioned when comparing it with previous studies.

- Table 3 should contain the sensitivity and specificity results.

- What are the limitations faced by the authors?

- What are the future works of this work.

- The results section should contain figures resulting from the system, such as the confusion matrix.

 Minor editing of English language required

Author Response

We are grateful to the reviewer for the encouraging remarks and insightful comments. In the light of their true or positive comments, we have improved our manuscript with the hope that it shall meet the high standards of your esteemed journal.

Reviewer 2 Report

The paper organized by the authors was good.

Authors need to clear that invasive or non-invasive method will give better accuracy.

Need more explanation for CT technologies.

Introduction part was very long. 

what is the new techniques implemented in the methodology. please highlight that in this study.

why did you t set the threshold point with help of rescaling intensity  ranges in between 92 to 98?

any specific reasons for d small-scale dataset?

so many networks are there why did you select the CRV-NET?. in that what all are the parameters you changed  for your proposed work.

List out the parameters presented in the U-Net model .

how can you concluded that your study showed better results in terms of  accuracy and robustness?

kindly i request authors to give the answers for these question.

Author Response

(The authors gave the same response as above.)

Reviewer 3 Report

Adjustment of accuracy and robustness in the recognition of Covid-19 is the main clinical need after the emergence of the new virus globally. Previous studies have made a fair contribution to the diagnosis of Covid-19, but the robustness factor is understood. This study provided a unique framework for accurate early diagnosis of Covid-19 with minimal computational cost that supports the robustness factor. This study proposed the CRV-NET architecture for robust recognition of Covid-19, which is a repurposed version of the generic U-NET architecture. A public dataset of Sars-Cov-2 CT scans was used for experimentation and preparation of a custom dataset. The custom dataset was used for training. The custom dataset is used to train the model with 331 ground truth images and 100 images are used to test the model. The accuracy of the proposed CRV-NET is 96.67%, which is better than the comparison with the generic U-NET architecture. CRV-NET's epoch value of seven is quite impressive and supports the robustness factor in terms of minimum computational cost. A comparison with the state of the art is performed to verify the accuracy and robustness factor and finally highlight the importance of CRV-NET in future perceptions. In the future, wearable gauges using the reliable federated learning scheme with an embedded part of CRV-NET will be used for early detection of Covid-19.

The article is of important innovation and touches sensitive topics with respect to the diagnosis of coronavirus-19 pathology and its continuous changes.

Attention should be paid to punctuation and paragraph errors.

We recommend reading and if appropriate citing the following articles:

PMID 34179691

PMID 34627709

Adjustment of accuracy and robustness in the recognition of Covid-19 is the main clinical need after the emergence of the new virus globally. Previous studies have made a fair contribution to the diagnosis of Covid-19, but the robustness factor is understood. This study provided a unique framework for accurate early diagnosis of Covid-19 with minimal computational cost that supports the robustness factor. This study proposed the CRV-NET architecture for robust recognition of Covid-19, which is a repurposed version of the generic U-NET architecture. A public dataset of Sars-Cov-2 CT scans was used for experimentation and preparation of a custom dataset. The custom dataset was used for training. The custom dataset is used to train the model with 331 ground truth images and 100 images are used to test the model. The accuracy of the proposed CRV-NET is 96.67%, which is better than the comparison with the generic U-NET architecture. CRV-NET's epoch value of seven is quite impressive and supports the robustness factor in terms of minimum computational cost. A comparison with the state of the art is performed to verify the accuracy and robustness factor and finally highlight the importance of CRV-NET in future perceptions. In the future, wearable gauges using the reliable federated learning scheme with an embedded part of CRV-NET will be used for early detection of Covid-19.

The article is of important innovation and touches sensitive topics with respect to the diagnosis of coronavirus-19 pathology and its continuous changes.

Attention should be paid to punctuation and paragraph errors.

We recommend reading and if appropriate citing the following articles:

PMID 34179691

PMID 34627709

Author Response

We are grateful to the reviewer for the encouraging remarks and insightful comments. In the light of their true or positive comments, we have improved our manuscript with the hope that it shall meet the high standards of your esteemed journal.

Concern #1

Adjustment of accuracy and robustness in the recognition of Covid-19 is the main clinical need after the emergence of the new virus globally. Previous studies have made a fair contribution to the diagnosis of Covid-19, but the robustness factor is understood. This study provided a unique framework for accurate early diagnosis of Covid-19 with minimal computational cost that supports the robustness factor. This study proposed the CRV-NET architecture for robust recognition of Covid-19, which is a repurposed version of the generic U-NET architecture. A public dataset of Sars-Cov-2 CT scans was used for experimentation and preparation of a custom dataset. The custom dataset was used for training. The custom dataset is used to train the model with 331 ground truth images and 100 images are used to test the model. The accuracy of the proposed CRV-NET is 96.67%, which is better than the comparison with the generic U-NET architecture. CRV-NET's epoch value of seven is quite impressive and supports the robustness factor in terms of minimum computational cost. A comparison with the state of the art is performed to verify the accuracy and robustness factor and finally highlight the importance of CRV-NET in future perceptions. In the future, wearable gauges using the reliable federated learning scheme with an embedded part of CRV-NET will be used for early detection of Covid-19.

Authors Response:

Respected reviewer, the highlighted concern raised is a valid and serious concern. Authors take this concern on serious note because it’s related to the presentation of the manuscript. In revised version of manuscript, the conclusion section is complete regenerated. Please review below changes

Adaptation of accuracy and robustness in recognition of Covid-19 is the top clinical requirement after the emergence of the new virus global. Previous studies delivered a fair contribution to the diagnosis of Covid-19 but the robustness factor is comprised. This research work proposed a unique structure of U-Net in terms of CRV-Net for accurate early diagnosis of Covid-19 with the least computational cost (least epoch ratio).. The proposed CRV-NET architecture worked for robust recognition of Covid-19 that is a refactored version of generic U-NET architecture. A public Sars-Cov-2 CT scan dataset is used for experimental purposes and customized dataset (derived from public Sars-Cov-2 CT scan dataset). The customized dataset is used for training the model with 331 ground truth images and 100 images are used for testing the model. Evaluation of proposed CRV-NET in terms of accuracy 96.67% is better than the comparison of generic U-NET architecture. Similarly, epoch value of CRV-NET is seven that is quite impressive and supportable for robustness factor (least computational cost).. Moreover, wearable gages will be used in the future for early diagnosis of Covid-19 by using the trustworthy federated learning work scheme with an embedded part of CRV-NET

Actual Changes:

In revised version of manuscript, the Conclusion section is completely regenerated

Round 2

Reviewer 1 Report

The authors treated the manuscript well.

Accept in present form.

Minor editing of English language required